# How the Analysis of the Pathogenetic Variants of DDR Genes Will Change the Management of Prostate Cancer Patients

**DOI:** 10.3390/ijms24010674

**Published:** 2022-12-30

**Authors:** Alessandro Sciarra, Marco Frisenda, Giulio Bevilacqua, Alessandro Gentilucci, Susanna Cattarino, Gianna Mariotti, Francesco Del Giudice, Giovanni Battista Di Pierro, Pietro Viscuso, Paolo Casale, Benjamin I. Chung, Riccardo Autorino, Simone Crivellaro, Stefano Salciccia

**Affiliations:** 1Department “Materno Infantile e Scienze Urologiche”, University Sapienza, 00161 Rome, Italy; 2Department of Urology, Humanitas Center, 20089 Milano, Italy; 3Department of Urology, University School of Stanford, Stanford, CA 94305, USA; 4Department of Urology, Rush University Medical Center, Chicago, IL 60612, USA; 5Department of Urology, University of Illinois Hospital, Chicago, IL 60612, USA

**Keywords:** prostate neoplasm, PARP inhibitor, *BRCA* gene, DDR gene, castration-resistant

## Abstract

Herein, we analyze answers achieved, open questions, and future perspectives regarding the analysis of the pathogenetic variants (PV) of DNA damage response (and repair) (DDR) genes in prostate cancer (PC) patients. The incidence of PVs in homologous recombination repair (HRR) genes among men with metastatic PC varied between 11% and 33%, which was significantly higher than that in non-metastatic PC, and *BRCA2* mutations were more frequent when compared to other DDR genes. The determination of the somatic or germline PVs of *BRCA2* was able to define a tailored therapy using PARP inhibitors in metastatic castration-resistant prostate cancer (mCRPC) progression after first-line therapy, with significant improvements in the radiologic progression-free survival (rPFS) and overall survival (OS) rates. We propose testing all metastatic PC patients for somatic and germline HRR mutations. Somatic determination on the primary site or on historic paraffin preparations with a temporal distance of no longer than 5 years should be preferred over metastatic site biopsies. The prognostic use of DDR PVs will also be used in selected high-risk cases with non-metastatic stages to better arrange controls and therapeutic primary options. We anticipate that the use of poly-ADP-ribose polymerase (PARP) inhibitors in hormone-sensitive prostate cancer (HSPC) and in combination with androgen receptor signaling inhibitors (ARSI) will be new strategies.

## 1. Premise

The systemic treatment of metastatic prostate cancer (mPC) has undergone an epochal positive evolution in recent years. In particular, this evolution has led to:-A shift from a clear distinction between the management of hormone-sensitive prostate cancer (HSPC) and the management of castration-resistant prostate cancer (CRPC) to a merger of the two phases. At present, the management of metastatic castration-resistant prostate cancer (mCRPC) represents a second line of therapy, sharing several of the recommended drugs for metastatic hormone-sensitive prostate cancer (mHSPC).-The concept of the anticipation of therapies as the main goal achieved, leading, in particular, to the use of both docetaxel and androgen receptor signaling inhibitors (ARSi) from the mCRPC to the mHSPC phase.

The point of failure of this process, which is accentuated due to the numerous therapeutic options that these patients have at their disposal, is the identification of valid prognostic indicators of therapy response and the construction of a tailored precision therapy for each specific patient.

DNA damage response (and repair) (DDR) genes are involved in the mechanisms of genetic instability, the repair of DNA aberrations during cell cycle, and the detection and repair of DNA damage, leading to apoptosis of dangerous mutated cells. Pathogenetic variants (PV) of DDR genes reduce the ability to effectively repair single- and double-strand breaks (SSB and DSB) of DNA damage. The poly-ADP-ribose polymerase (PARP) system is involved in detecting and repairing primarily SSBs, whereas the homologous recombination repair (HRR) pathway of DDR genes is primarily involved in repairing DSB DNA lesions. *BRCA1-2* is the most famous gene involved in the HRR system, and its PVs are associated with an increased risk of developing breast, ovarian, prostatic, pancreatic, and colon cancers [1].

*BRCA* PVs, in several neoplasms, have been associated with:-Short metastatic-free survival-Short cancer-specific survival-The prediction of responses to PARP inhibitors and to platinum salts

## 2. DDR Analysis: Prognostic Role and Prediction of Response to Therapy

### 2.1. Incidence

Several studies and registers have reported the incidence of PVs of different DDR genes in PC patients. The incidence of germline PVs in DDR genes among men with metastatic prostate cancer (mPC) varied between 11% and 33%, which was significantly higher than that in non-metastatic prostate cancer (nmPC) (5–10%) [2]. *BRCA2* PVs were more frequent (5.3%) when compared to *CHEK2* (1.9%), *ATM* (1.6%), *BRCA1* (0.9%), and *RAD51* (0.4%). In mCRPC, the incidence of somatic DDR mutations was 24%, with a *BRCA2* incidence of 13%, followed by an *ATM* incidence of 7.3%, an *MSH2* incidence of 2.0%, and a *BRCA1* incidence of 0.3% [3].

The International Stand Up of Cancer Register, which analyzes mCRPC, showed a 23% incidence of somatic DDR PVs (in particular, a *BRCA2* of 13%, *ATM* of 7.3%, *MSH2* of 2%, and *BRCA1*-*RAD51* of 0.3%) [4,5].

The Profound study [6], which evaluated 2792 biopsies of mCRPC, showed DDR PVs in 28% of the cases, with a similar incidence considering the primary tumor (27%) or metastatic sites (32%). Similar data were also observed in a Chinese population (326 cases with mPC or localized high-risk PC), in which 9.8% of the analyzed samples (95% confidence interval [CI]: 6.5–13%) carried the following PV incidence rates: *BRCA2* of 6.3%, *BRCA1* of 0.63%, *ATM* of 0.63%, and others of 2.5%, which suggested a relatively high incidence of DDR genes PVs independent of ethnicity [7].

### 2.2. Prognostic Role in Non-Metastatic PC

The determination of somatic DDR PVs in nmPC could have an impact on the oncological outcomes of these patients. Data regarding the implications of germline and somatic DDR defects in early prostate cancer are needed to tailor treatments with a precision medicine perspective (Table 1).

In this context, the IMPACT study analyzed the role of PSA screening in *BRCA* PV carriers compared to controls after 3 years of follow-up [8]. Differences between the carriers and controls were always higher, considering the analysis of *BRCA2* over *BRCA1*. The incidence of PC at biopsy was 5.2% in *BRCA2* PV carriers and 3.0% in non-carriers. The incidence of high or intermediate risk in PC cases changed from 77% in *BRCA2* carriers to 40% in non-carriers. Similarly, the ISUP grade 1 distribution changed from 73% in non-carriers to 37% in *BRCA2* carriers. The results of this study suggest that *BRCA* PV carriers may harbor a more aggressive disease that we need to consider when selecting a treatment.

Carter H.B. et al. evaluated whether germline PVs were associated with grade reclassification (GR) in patients undergoing active surveillance (AS) [9]. Analyzing 1211 PC patients selected for AS, the incidence of tumor staging upgrade at 2, 5, and 10 years was 27%, 50%, and 78%, respectively, in *BRCA2* PV carriers and 10%, 22%, and 40%, respectively, in non-carriers. However, given the retrospective nature of this study and the absence of multiparametric magnetic resonance imaging (mMRI) or targeted biopsies, the exact role of active surveillance in patients carrying germline PVs remains controversial and needs to be evaluated in prospective studies.

The prognostic role of HRR PVs in localized disease was also evaluated in the specific stetting of active treatments, such as surgery and radiotherapy. Castro E. et al. evaluated the effect of *BRCA* PVs on metastatic relapse and cause-specific survival after radical treatments (surgery and radiotherapy) for localized PC [10]. At 3, 5, and 10 years after treatment, 97%, 94%, and 84%, respectively, of non-carriers and 90%, 72%, and 50%, respectively, of carriers were free from metastasis (*p* < 0.001). The 3-, 5-, and 10-year CSS rates were significantly better in the non-carrier cohort (99%, 97%, and 85%, respectively) than in the carriers (96%, 76%, and 61%, respectively; *p* < 0.001). Multivariate analysis confirmed *BRCA* PVs as an independent prognostic factor for metastasis-free survival (MFS) (hazard ratio [HR]: 2.36; 95% confidence interval [CI], 1.38–4.03; *p* = 0.002) and cancer-specific survival (CSS) (HR: 2.17; 95% CI: 1.16–4.07; *p* = 0.016). Similar results were also observed by Martinez Chanza M. et al. in a retrospective series of 380 patients with localized and metastatic hormone-sensitive PC [11], concluding that *BRCA* PVs were associated with a greater relapse risk in localized disease. Moreover, Castro E. et al., in a retrospective study involving 2019 patients with PC, found that *BRCA1-2* PVs were more frequently associated with a Gleason score of ≥ 8 (*p* = 0.00003), T3/T4 stage (*p* = 0.003), nodal involvement (*p* = 0.00005), and metastases at diagnosis (*p* = 0.005) [12].

### 2.3. Prognostic Role in Metastatic PC

Several studies have explored the prognostic role of DDR PVs in mCRPC patients treated with standard therapies (Table 1). Annala M. et al., in a retrospective study of 319 patients with mCRPC, showed that DDR PVs carriers had a significantly shorter progression-free survival (PFS) rate than that of non-carriers (3.3 vs. 6.2 months, *p* = 0.01) when treated with ARSI as first-line therapy [13]. Contrarily, Antonarakis et al., in a study of 172 mCRPC patients receiving first-line ARSI, observed that *ATM-BRCA1/2* carriers trended towards longer PFS than did noncarriers (15 vs. 10.8 months, *p* = 0.090) [14].

Prorepair-B was the first prospective trial to analyze the prognostic impact of *BRCA1–2* and other DDR genes on CSS in mCRPC patients [15]. Considering all DDR defects together, the study failed to find significant differences in CSS rates (DDR PV carriers, 23.3 months vs. non-carriers, 33.2 months; *p* = 0.264; HR: 1–32: 95% CI: 0.81–2.17) between carriers and non-carriers. On the contrary, *BRCA2* PVs alone resulted in an independent prognostic factor negatively affecting CSS rates (*BRCA2* PVs carriers, 17.4 months versus non-carriers, 33.2 months; *p* = 0.027; HR: 2.10; 95% CI: 1.07–4.10). In a subgroup analysis, *BRCA2* PVs predicted a shorter CSS rate in mCRPC when treated with a sequence docetaxel-ARSI (10.7 months) than with a sequence ARSI-docetaxel (24.0 months). The results of this study suggested that *BRCA2* PVs have a deleterious impact on outcomes in mCRPC patients related to the choice of first-line therapy.

## 3. DDR Analysis: How to Perform

Some DDR alterations may occur early in the evolution of aggressive tumors and could be detected by a prostate biopsy or prostatectomy. Other events may occur later during progression and a metastatic tumor biopsy could be the preferred approach. However, biopsies of metastatic lesions often could be dangerous or not feasible.

### 3.1. Recent vs. Archived Somatic Samples

In the PROfound study, a total of 4858 tissue samples were tested and reported centrally [6]. Next generation sequencing (NGS) results were obtained in 58% of the samples (69% of patients). Considering all the samples obtained, 83% were primary tumor samples (of which 96% were archival and 4% were newly obtained) and 17% were metastatic tumor samples (of which 67% were archival and 33% were newly obtained). NGS results were generated more frequently from newly obtained samples compared with archival samples (63.9% vs. 56.9%, respectively) and from metastatic samples compared with primary samples (63.9% vs. 56.2%, respectively). The authors observed that the DNA yield affected the NGS results (area under the curve (AUC) = 0.6292) more than other variables did (such as sample age or percentage of tumor content). Although the generation of NGS results declined with increasing sample ages, approximately 50% of the samples aged >10 years generated results. Another parameter to be considered for eligibility is total tissue volume. In the PROfound study [6], the proportion of samples generating an NGS result was higher in samples with a tissue volume of >0.6 mm^3^ (58.9%; 95% CI: 57.4–60.3) than in samples from the 0.2 and ≤ 0.6 mm^3^ cohorts (32.1%; 95% CI: 24.4–40.6). Finally, the PROfound study showed that tissue testing to identify HRR alterations is feasible and that high-quality tumor tissue samples are key for obtaining NGS results and identifying patients with mCRPC who may benefit from the olaparib treatment.

### 3.2. Somatic vs. Germline Analysis in PC

The somatic analysis of DDR PVs in PC should first be considered when comparing the DDR PVs to a germline analysis. When mutations of DDR genes are acquired during the progression of the disease, a biopsy of the metastatic tumor represents the ideal approach to identify molecular alterations. However, biopsies of metastatic lesions can be challenging or not feasible, and at the same time, a single biopsy may not reveal tumor heterogeneity among metastases. The PROfound study also pointed out that 30% of biopsy samples may not be of sufficient quality for gene sequencing [6]. Most metastatic lesions in prostate cancer occur in bone and obtaining a biopsy sample from bone metastases could be particularly difficult both for the patient and the operator due to the invasiveness of the procedure, the need for anesthesia, costs, and post-operative pain. Furthermore, processing bone biopsy samples, which requires decalcification, can lead to a reduction in the quantity and quality of the DNA. In the PROfound study, NGS tests on bone samples had lower proportions of the results when compared to all the other sampled sites (42.6%) [6].

Germline analysis is linked to a genetic counseling for PC patients and their relatives. Pre-test consultations are intended to collect data and the family history of > 3 generations on both sides of the family, discussing genetic testing options and explaining potential genetic tests. Post-test consultations should discuss and explain genetic test results, potential predictions on the risk for different neoplasms, and indications for possible intensive screening strategies. Germline testing can be developed through different approaches using small/focused panels (5–6 genes), cancer-specific panels (10–15 genes), or large, comprehensive panels (≥ 80 genes) [16].

The American College of Medical Genetics and Genomics (ACMG) investigates the clinical significance of sequence variants and creates a guideline for the use of standard terminology [17,18]. According to ACMG criteria, the results of a test can be divided into five classes:-Non-pathogenetic variants (probability of pathogenicity (PP) of < 0.001)-Likely not pathogenic or of little clinical significance (PP: 0.001–0.049)-Uncertain significance (PP: 0.05–0.949)-Likely pathogenic (PP: 0.95–0.99)-Pathogenic (PP of > 0.99).

If a test offers no results or provides as a result a “variant of uncertain significance” (VUS), this should not be classified as a pathogenic variant for the disease, and more often, VUSs are eventually considered as having neutral/low clinical significance [19].

The Philadelphia PC consensus conference [16], regarding germline analysis in PC patients, stated:-test all metastatic PCs (either hormone-sensitive (HS) or castration-resistant (CR))-test all PCs with a significant family history of PC or hereditary breast and ovarian cancer syndrome and Lynch syndrome-in PC cases with somatic mutations, perform germline evaluations in all related family members-in nmPC, use the reflex test with an initial analysis of priority genes, followed by expanded panels with a particular focus on *BRCA2*

When a constitutional variant occurs, in addition to the possibility of accessing possible treatment with a PARP inhibitor, the patient will be able to have access to a preventive path for healthy carriers (through the launch of clinical surveillance programs, which are instrumental, or the implementation of risk reduction strategies) through oncogenetic counseling.

### 3.3. cDNA

The analysis of free circulating DNA (cDNA) is a promising approach as it may overcome the difficulties that are, in many cases, associated with obtaining tissue; currently, however, there are no data that allow this test to be used reliably. The first study that analyzed cDNA in this field was the GHALAND study [20]. It was a phase two study that investigated treatment outcomes with Niraparib in mCRPC patients with DDR PVs. The treatment efficacy analysis was performed on the amount of circulating tumor cells (CTC) reported from the eighth week of treatment. The best results were obtained in the *BRCA* PVs cohort over the non-*BRCA* PVs cohort, with 24% of the CTC response. However, the current commercial tests for the study of cDNA show conflicting results, with inconsistencies reaching 40% of analyses and a risk of inappropriate or missing treatments for patients [21].

## 4. PARP Inhibitors as Tailored Therapy

HRR PVs in tumor cells induce a dependency for single-strand-break reparation through the poly-adenosine diphosphate-ribose-polymerase (PARP) system, providing the rationale for developing PARP inhibitors. The PARP system is involved in DNA repair via homologous recombination in HRR-deficient cells. PARP inhibitors block the PARP system, causing the accumulation of DNA damages in DDR-defective tumor cells.

Five prospective clinical trials have made it possible to clarify the role of HRR and the PARP system in PC, clarify their prognostic value, and obtain a tailored therapy for the treatment of mCRPC that progresses after first-line treatment with ARSi and taxanes.

The TOPARP-B phase II trial [22] has confirmed the antitumoral activity of olaparib in mCRPC, with specific DDR PVs. Both drug doses (300 mg vs. 400 mg) and the specific type of DDR PVs may influence anti-tumor activity, with the greatest results in the subgroup with germline or somatic *BRCA 1-2* alterations as compared to *ATM* or *CDK 12* aberrations.

The overall antitumor activity of olaparib in patients with DDR PVs by gene subgroup shows the PSA-50% responses in *BRCA 1-2*, *ATM*, and *CDK12* at 76.7%, 5.3%, and 0%, respectively, and the CTC conversion at 77.3%, 50%, and 41.7%, respectively.

The PROFOUND [6] phase three trial analyzed olaparib in two different cohorts based on the type of DDR PVs. It demonstrated a significant advantage of olaparib on ARSi only in cohort A (*BRCA1-2* or *ATM* defect) and, in particular, in *BRCA2* PVs, either in terms of radiologic progression-free survival (rPFS) (7.4 months versus 3.5 months; hazard ratio (HR) of 0.34; 95% CI: 0.25–0.47; *p* < 0.001) or overall survival (OS) (19.1 months versus 14.7 months, HR 0.69, 95% CI: 0.50–0.97, *p* = 0.02).

The TRITON-2 [23] phase two trial analyzed rucaparib according to HRR defects; 115 patients had a *BRCA1-2* alteration (44 germline and 71 somatic). The objective response rate (ORR) was 43.5% (95% CI: 38.1–63.4%), and no differences were observed between the germline and somatic PVs patients on a median follow-up of 17.1 months. The PSA response rate was 54.8% (95% CI: 45.2–64.1%) and the PSA responses appeared smaller in the *BRCA1* (15.4%) or mono-allelic patients (11.1%) compared to the *BRCA2* (59.8%) or biallelic patients (75.0%). Seventy-eight patients had a non-*BRCA* HRR alteration, and the PSA response rates were 4.1%, 6.7%, and 16.7% in the *ATM* group (49 patients), *CDK12* cohort (15 patients), and *CHEK2* group (12 patients), respectively. The ORR was 10.5% in the *ATM* group, 0% in the *CDK12* cohort, and 11.1% in the *CHEK2* group.

The GALAHAD [20] phase two trial selected cases on the basis of HRR defects for treatment with miraparib. The ORR was 41% (95% CI: 23.5–61.1%) in cases with *BRCA* defects and 9% (95% CI: 1.1–29.2%) in cases with other HRR PVs. The median PFS and OS in the *BRCA* PVs were 8.2 months (95% CI: 5.2–11.1 months) and 12.6 months (95% CI: 9.2–15.7 months), respectively, vs. 5.3 months (95% CI: 1.9–5.7 months) and 14.0 months (95% CI: 5.3–20.1 months), respectively, in the non-*BRCA* cases.

The TALAPRO1 [24] phase two trial showed a 29.8% ORR (95% CI: 21.2–39—6%) associated with talazoparib (46% in cases with *BRCA1-2* PVs with an rPFS of 11.2 months).

The overall antitumor activity of talazoparib produced PSA-50% responses in the *BRCA 1-2*, *ATM,* and other PVs patients of 66%, 7%, and 6%, respectively, and the CTC conversion rates were 81%, 50%, and 20%, respectively.

## 5. Answers Achieved, Open Questions, and Future Perspectives

### 5.1. Answers Achieved

In particular, the following answers have been achieved:-The incidence of germline PVs in HRR genes among men with mPC varies between 11% and 33%, which is significantly higher than that of nmPC, and the *BRCA2* PVs were more frequent when compared to other HRRs.-The determination of somatic or germline HRR PVs and, in particular, *BRCA2*, is able to define a tailored therapy with PARP inhibitors in mCRPC that progresses after first-line therapy, with significant improvements in rPFS and OS. This point reached recommendations from international guidelines and approval from the FDA and EMA for olaparib and rucaparib.

### 5.2. Open Questions

Some relevant open questions remain, primarily on the basis of the methodology of HRR PVs determination (Figure 1):


-Conflicting results remain when comparing the somatic determination of HRRs between the primary site from prostate biopsy or prostatectomy and the biopsy on metastases (some studies found an incidence of 10% in primary tumors and 27% in metastatic samples, whereas others found similar detection rates between prostate biopsies (27%) and those of metastases (32%)) [13,14,15].-Somatic determination on a metastatic site, in particular, bone, may be associated with various biases (in 30% of cases), as well as possible side effects. On the other hand, the de novo determination on the prostate is not always possible at the time of the diagnosis of metastatic disease. The reliability of the analysis on paraffin preparations is conditioned by the temporal distance (good if <5 years and bad if >10 years).-Data on the use of circulating free DNA are still incomplete.


### 5.3. Future Perspectives

The IMPACT study [8] evaluated a screening strategy in cases of *BRCA* PVs, allowing for information on how to manage the patient based on HRR determination, even in the early stages of the disease.

A point of great interest is the crosstalk between androgen receptors (AR) and DNA repair. The PARP system is involved in androgen-dependent transcription, and PARP inhibitors can impair it. On the other hand, the AR pathway regulates the transcription of DNA repair genes and androgen depletion or AR signaling inhibitors, which impair HRR, which rends a tumor susceptible to the PARP inhibitors. In PC xenograft experiments, a better effect was reported when using enzalutamide prior to olaparib compared to the use of single monotherapies [25]. The PROpel trial [26] showed that the combination of abiraterone and olaparib in mCRPC patients as first-line in non-selected cases, based on HRR, is able to produce a 39% reduction in radiologic progression or death. The MAGNITUDE study [27] analyzed the combination of abiraterone with niraparib in mCRPC and HRR defect patients, with initial conflicting results based on the different HRR gene PVs.

In particular, for the future, it can be suggested that:-A geneticist will be included in our multidisciplinary groups on PC.-The prognostic value of HRR PVs will also be used in selected high-risk cases with non-metastatic stages to better arrange controls and therapeutic primary options.-The role of HRR genes other than *BRCA2* will be better characterized, and new tailored therapies will be considered on the basis of defects other than *BRCA*.-The anticipation in the use of PARP inhibitors (HSPC) will be investigated.-The crosstalk between AR and DNA repair will be used to conduce to new combination strategies using ARSi plus PARP inhibitors or better therapeutic sequences (Table 2).

## 6. Conclusions

There are some differences in the actual recommendations given by the guidelines of the various international societies. To date, the use of DDR gene analysis and PARP inhibitor therapy could be summarized as follows, including the prognostic, therapeutic, and genetic counseling potential (Figure 2):


-Test all mPC patients (either hormone-sensitive or castration-resistant) for somatic and germline HRR PVs. The use of this analysis in high-risk nmPC patients or for those with a significant family history of PC must be carefully discussed.-Use both germline and somatic tests and large gene panels with priority for *BRCA2*, leaving the determination of circulating free DNA in cases in which the somatic or germline determination is not suitable or reliable. Somatic determination on the primary site (prostate, when available) or on historic paraffin preparations with a temporal distance of no longer than 5 years should be preferred to metastatic site biopsies. The analysis must be performed in a specialized laboratory, following the criteria for evidence-based networks for the interpretation of PVs [16].-In mPC cases with somatic and germline *BRCA* PVs, offer a germline evaluation for all related family members after genetic counseling so as to conduct early cancer screening. The same germline evaluation is suggested in cases with high-risk nmPC patients with two or more relatives with a history of mammary, ovarian, or prostate neoplasms, or a history of early prostate cancer development (<55 years).-Consider a PARP inhibitor tailored therapy in cases with mCRPC that progresses after first-line therapy and that shows somatic and/or germline *BRCA2* PVs.


## Figures and Tables

**Figure 1 ijms-24-00674-f001:**
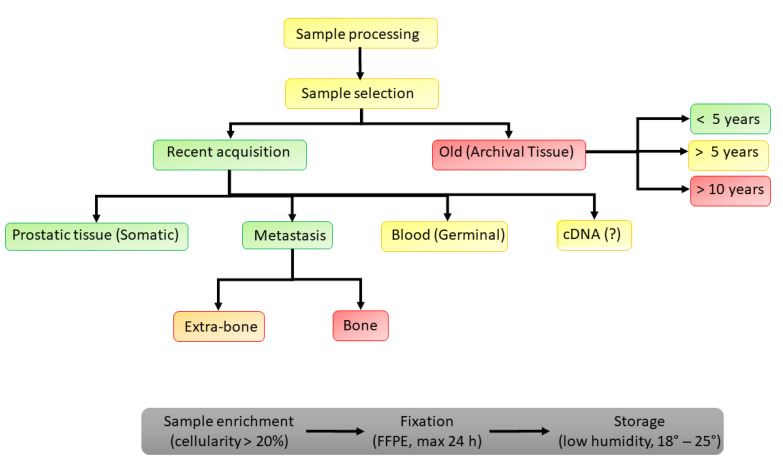
How to detect HRR pathogenetic variants in PC patients (FFPE, formalin-fixed paraffin-embedded).

**Figure 2 ijms-24-00674-f002:**
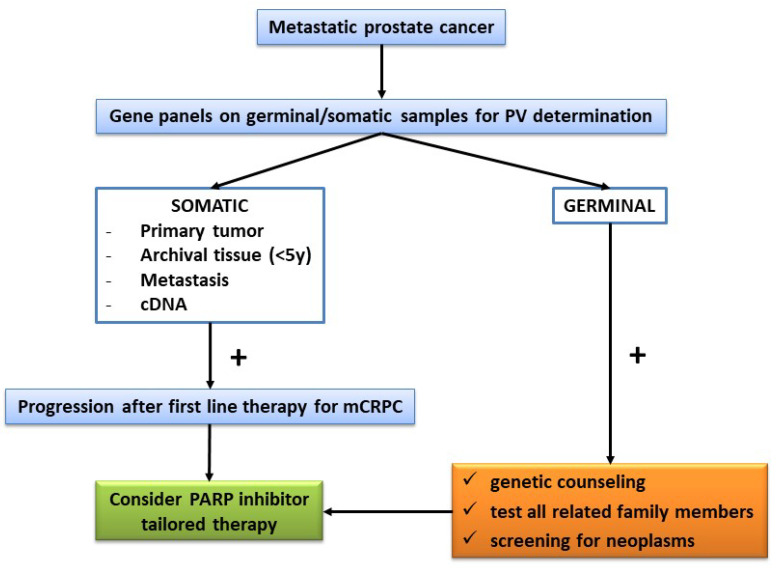
Flow chart on the use of HRR determination and PARP inhibitor strategy in metastatic prostate cancer patients.

**Table 1 ijms-24-00674-t001:** Studies reporting the outcomes on the prognostic role of DDR PVs in localized PC and mPC (csPC, clinical significant prostate cancer; MFS, metastasis-free survival; CSS, cause-specific survival; EFS, event-free survival, defined as a prostate-specific antigen (PSA) of > 2.0 ng/mL in patients treated with radiotherapy and a PSA of > 0.2 ng/mL in patients treated with radical prostatectomy, metastasis, or death).

Study	Study Design	Ethnicity	Incidence of DDR Gene PVs	Population	Outcomes Analyzed	Oncological Results in Carriers vs. No Carriers
Localized PC						
Page E.C. et al., 2019. IMPACT study [8]	Prospective	Caucasian	Not applicable	PSA screening 902 *BRCA-2* carriers 497 *BRCA-2* no carriers	- PC incidence (rate per 1000 persons per year) -csPC	(19.4% vs. 12.0%; *p* = 0.03 (77% vs. 40%; *p* = 0.01)
Carter H.B. et al., 2019 [9]	Retrospective	Caucasian	All genes (*BRCA1-2* plus *ATM*) - any upgrading (3.81%) - upgraded GG1 to ≥ 3 (5.10%) - upgraded GG1 to ≥ 2 (3.89%)	Active surveillance - 1211 patients	grade reclassification (GR) at 2, 5 and 10 years	(27%, 50% and 78% vs. 10%, 22% and 40%)
Castro E. et al., 2015 [10]	Retrospective	Caucasian	*BRCA 1-2* (5.15)	1302 patients with local or locally advanced PC treated with RT or RP - 67 carriers - 1235 no carriers	MFS at 3, 5, and 10 years CSS at 3.5 and 10 years	(90%, 72% and 50% vs. 97%, 94% and 84%, *p* < 0.001) (96%, 76% and 61% vs. 99%, 97% and 85%, *p* < 0.001)
Martinez Chanza N. et al., 2022 [11]	Retrospective	Caucasian	3.5%	258 patients with localized PC treated with RP or RT - 8 carriers - 250 no carriers	EFS MFS	18.1 vs. 57 months (HR 1.73; 95% CI: 0.63–4.74; *p* = 0.28) 37 vs. 153 months (HR 2.77; 95% CI: 0.84–9.14; *p* = 0.08)
Castro E. et al., 2013 [12]	Retrospective	Caucasian	*BRCA1-2* (3.81%)	2019 patients at diagnosis - 79 carriers - 1940 no carriers	MFS CCS OS N+	(77% *v* 93%; *p* = 0.0001) (82% *v* 96%; *p* = 9 × 10^−8^) (8.1 vs. 12.9 years; *p* = 1 × 10^−7^) (N1: 15% *v* 5%; *p* = 0.0005)
Metastatic PC						
Annala M. et al., 2017 [13]	Prospective	Caucasian	7.5%	133 patients with mCRPC treated with first-line therapy with docetaxel, abiraterone, or enzalutamide - 22 carriers - 113 no carriers	PFS on first-line AR target therapy PFS on first-line docetaxel	3.3 months (95% CI: 2.7–3.9) vs. 6.2 (95% CI: 5.1–7.3), *p* = 0.01 7.2 months (95% CI: 5.6–8.7) vs. 8.0 (95% CI: 7.1–9.1), *p* = 0.127
Antonarakis E. et al., 2018 [14]	Prospective	Caucasian	12.8%	172 patients with mCRPC treated with first-line therapy with abiraterone or enzalutamide - 22 carriers - 152 no carriers	PSA response rate PSA-PFS PFS OS	77% vs. 59% (*p* = 0.158) median 10.2 vs. 7.6 months; [HR] 0.64, 95% CI: 0.39–1.04; *p* = 0.070 median 13.3 vs. 10.3 months; HR 0.67, 95% CI: 0.41–1.09; *p* = 0.107 median 41.1 vs. 28.3 months; HR 0.58, 95% CI: 0.30–1.11; *p* = 0.097
Castro E. et al., 2019 PROREPAIR-B [15]	Prospective	Caucasian	16.2%	419 patients with mCRPC treated with first-line therapy with abiraterone or enzalutamide - 68 carriers - 351 no carriers	Time to mCRPC CCS CCS (only *BRCA2*)	22.8 vs. 28.4 months; *p* = 0.007 23.3 vs. 33.2 months; HR, 1.32; 95% CI, 0.81 to 2.17, *p* = 0.264 17.4 vs. 33.2 months; HR, 2.10; 95% CI, 1.07 to 4.10; *p* = 0.027

**Table 2 ijms-24-00674-t002:** Answers achieved and future perspectives regarding DDR gene analysis in prostate cancer. PV, pathogenetic variants; mPC, metastatic prostate cancer; nmPC, non-metastatic prostate cancer; mCRPC, metastatic castration-resistant prostate cancer; HRR, homologous recombination repair; mHSPC, metastatic hormone-sensitive prostate cancer.

Answers Achieved	Future Perspectives
Incidence of PVs in HRR genes = 11–33%, higher in mPC than in nmPC	Include a geneticist in multidisciplinary groups on PC
*BRCA2* PVs more frequent than other HRRs	Use of the prognostic analysis of HRR PVs in selected high-risk cases with nmPC
Somatic or germline HRR PVs define tailored therapy with PARP inhibitors in mCRPC	Anticipation in the use of PARP inhibitors from mCRPC to mHSPC
Somatic determination on primary site with a temporal distance of less than 5 years as preferred for HRR analysis	New combination strategies using ARSi plus PARP inhibitors

## Data Availability

All data are available on academic research databases (PubMed, Medline, Web of Science, Scopus and the Cochrane library).

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
