# Peer review of "How the Analysis of the Pathogenetic Variants of DDR Genes Will Change the Management of Prostate Cancer Patients"

_ijms, 2022, doi:10.3390/ijms24010674_

Round 1
Reviewer 1 Report
The manuscript from Sciarra et al. is focused on a relevant topic regarding Prostate cancer, i.e. the role of genetic data in the overall management of the patients.
Nevertheless, the paper would have benefit from the contribution of an expert in medical genetics due to a low accuracy in the description of some data/nomenclature or the general architecture of some paragraphs that undermines the value and the readability of the paper.
Some observations:
-line 44 “DDR mutations in pathogenic variants” is nonsense
-line 60: the name of the cited gene is CHEK2, not CHECK2
-the name of the genes are usually reported in italics to differentiate them from the coded protein and avoid confusion among genes and protein. This issue is present throughout the manuscript.
-it is confounding, in particular for a casual reader, to alternative use pathogenic variants or mutations.
While the term mutation is increasingly less used, it could be clearer to use pathogenic variant while referring to a germline variant and mutation to a somatic variant. Anyway, germline and somatic are just specific.
-line 74: independent of, not independent to
-line 88: the term carrier is usually adopted for germline variant carriers, hence the use of “mutation” is not clear.
-the paragraph 2.3 show the same problems above, i.e the use of “mutation” and “carrier” while referring to germline data. The content of the paragraph could be clearly higlighted just from the title.
-line 176: counseling vs line 321: counselling
-line 182-183: references? Some widely used commercial panels include 72 genes.
-line 183: the sentence is convoluted
-line 186: “classified” is the correct term instead of “considered”. Reference for “more often VUS are eventually considered as non-pathogenic”?
-line 187: the ENIGMA classification is not specific of ENIGMA but ACMG derived. Besides that, variant classification could be anticipated in the text.
-line 224-231: formatting issue
Author Response
We thank Reviewers for the positive comments and constructive requests. We answered to all their requests so to improve the quality of our analysis
REVIEWER 1
The manuscript from Sciarra et al. is focused on a relevant topic regarding Prostate cancer, i.e. the role of genetic data in the overall management of the patients.
Nevertheless, the paper would have benefit from the contribution of an expert in medical genetics due to a low accuracy in the description of some data/nomenclature or the general architecture of some paragraphs that undermines the value and the readability of the paper.
Some observations:
Q1: line 44 “DDR mutations in pathogenic variants” is nonsense
A1: The line 48 has been revised in “Pathogenetic variants of DDR genes mutations reduce the ability to effectively repair single and double-strand breaks (SSB and DSB) DNA damage”
Q2: line 60: the name of the cited gene is CHEK2, not CHECK2
A2. The line 66 has been revised in CHEK2.
Q3: the name of the genes are usually reported in italics to differentiate them from the coded protein and avoid confusion among genes and protein. This issue is present throughout the manuscript.
A3: All the names of the genes have been modified and reported in italics.
Q4: it is confounding, in particular for a casual reader, to alternative use pathogenic variants or mutations. While the term mutation is increasingly less used, it could be clearer to use pathogenic variant while referring to a germline variant and mutation to a somatic variant. Anyway, germline and somatic are just specific.
A4: all the terms associated to the germline variants have been changed to “Pathogenic variants” in the text.
Q5: line 74: independent of, not independent to
A5: revised in “independent of” (line 77)
Q6: line 88: the term carrier is usually adopted for germline variant carriers, hence the use of “mutation” is not clear.
Q6: Revised in terminology (lines 92-93)
Q7: the paragraph 2.3 shows the same problems above, i.e the use of “mutation” and “carrier” while referring to germline data. The content of the paragraph could be clearly higlighted just from the title.
Q7: The paragraph 2.3 has been revised and the title clarified.
Q8: line 176: counseling vs line 321: counselling
Q8: changed all in counseling
Q9: line 182-183: references? Some widely used commercial panels include 72 genes
A9: The reference has been add to the paragraph (line 188).
Q10: line 183: the sentence is convoluted
A10: the sentence has been removed
Q11: line 186: “classified” is the correct term instead of “considered”. Reference for “more often VUS are eventually considered as non-pathogenic”?
Q11: line 200: term “considered” modified in “classified”. Added reference at line 201.
Q12: line 187: the ENIGMA classification is not specific of ENIGMA but ACMG derived. Besides that, variant classification could be anticipated in the text.
Q12: The sentence “ENIGMA criteria derived from the American College of Medical Genetics and Genomics (ACMG) guidelines for interpretation of sequence variations.” Has been add to the text (lines 191-193) and anticipated.
Q13: line 224-231: formatting issue
Q13: the Form of the text has been revised (line 231-236).
Reviewer 2 Report
Sciarra et al. reviewed about the importance of mutations in DNA damage response (DDR) genes in prostate cancer for the management of patients. Pathogenic variants (PV) in DDR gene (especially BRCA2) are found in metastatic prostate cancer (mPC) more frequently than non-metastatic prostate cancer. Since PV in homologous recombination repair (HRR) genes including BRCA2 is a prediction marker for sensitivity of PARP inhibitor, testing mPC patients for HRR gene mutation could find responder to PARP inhibitor and extend their survival time with the treatment of PAPR inhibitor. This is somehow interesting, but I do not agree to publish this review in present form.
Comments
1. There are few citations in the background information of prostate cancer (section 1). The authors should cite in the sections about metastasis of prostate cancer and DDR mutations in prostate cancer.
2. The strategy of PARP inhibitor treatment to mPC patients is previously reviewed and seems not very informative. I could not catch the important point.
3. Describe the abbreviations when they firstly appear.
4. Gene names should be italic.
5. The English language and style are very confusing and hard to understand especially the statistics in section 2.
Author Response
We thank Reviewers for the positive comments and constructive requests. We answered to all their requests so to improve the quality of our analysis
REVIEWER 2:
Q1: There are few citations in the background information of prostate cancer (section 1). The authors should cite in the sections about metastasis of prostate cancer and DDR mutations in prostate cancer.
A1: Considering limitation in words count, we preferred to immediately arrive to the aim of this review. A long introduction describing prostate cancer and DDR characteristics is out of our aim.
Q2: The strategy of PARP inhibitor treatment to mPC patients is previously reviewed and seems not very informative. I could not catch the important point.
A2: We absolutely agree with your comment. The aim of our review is to specify why DDR genetic analysis can really be different and successful than previous analysis. As stated in the review, the strength of DDR analysis is the prognostic power, the prediction on treatment response and the possibility to have a really tailored therapy based on genetic analysis with PARP inhibitor. In this view, we believe that and synthetic description of PARP inhibitor trials and results can complete our aim, differently than previous reviews.
Q3: Describe the abbreviations when they firstly appear.
A3: all the abbreviations have been fixed
Q4: Gene names should be italic.
A4: Revised
Q5: The English language and style are very confusing and hard to understand especially the statistics in section 2.
A5: English language has been extensively improved in all section by an English native colleague.
Reviewer 3 Report
The authors have provided a timely review focused on DDR and HRR pathogenic variables in selected pCa patient clinical trials. The prognostic value of DDR gene mutations in 5 local PCa and 3 mCPRC trials as well as clinical trials using PARP inhibitors as tailored / precision therapy.
They also discuss issues and pitfalls of methods used to determine patient DDR gene pathologic variants including sample viability for NGS, issues of somatic vs germline patient tumor DDR PV analysis, and blood based biomarkers, cDNA and CTCs.
For section 5.0 of their manuscript, I would suggest highlighting the AR / PARP crosstalk, AR transcription of DDR genes and AR blockade in HRR gene expression. Also, place the answers, open questions and suggestions in individual tables or other format with added references (if possible) to highlight the authors view of the current / future status of DDR and HRR targets in PCa.
Check spelling: line 315 "to conduce to now combinations" vs to conduce new combinations"
The figures, tables and schemes are very helpful. Table 1. contains the trial reference and an understandable summary of the data that supports the manuscript. The conclusions offer a flow chart summarizing the arguments presented. This review is clear and relevant summary of PCa clinical trials covering DDR gene PVs.
Author Response
We thank Reviewers for the positive comments and constructive requests. We answered to all their requests so to improve the quality of our analysis
REVIEWER 3
Q1: For section 5.0 of their manuscript, I would suggest highlighting the AR / PARP crosstalk, AR transcription of DDR genes and AR blockade in HRR gene expression. Also, place the answers, open questions and suggestions in individual tables or other format with added references (if possible) to highlight the authors view of the current / future status of DDR and HRR targets in PCa.
A1: We agree with you that the crosstalk between AR and PARP is of great interest. We increase the description on lines 304-314 and reference 25. A new table has been created in order to re-organize the topics of chapter 5, as suggested.
Q2: Check spelling: line 315 "to conduce to now combinations" vs to conduce new combinations"
A2: Revised “now” in “new”.
Round 2
Reviewer 1 Report
The revised version of the manuscript still presents some inaccuracies.
Line 15: DDR is the acronym for DNA Damage Response (and repair). Check this also throughout the manuscript (e.g. line 46). It seems the authors are not so familiar with molecular genetics but these acronyms are of common and international use. See also the comment about ENIGMA consortium.
Line 19: the genes name has to be written in italics. Check throughout the manuscript.
Line 48 vs Line 53: the abbreviation was just established. Correct line 53.
Line 65: the paragraph is at risk of to be confounding. The term mutations is used as referred to germline or somatic variants? Germline or somatic are quite specific and variants is a better term instead of mutation.
Line 181: what does it mean "genetic counseling in the family"? The genetic counseling is offered to the proband/patients. Rephrase e.g genetic counseling for PC patients.
Line 189: the ENIGMA criteria are more complex than as stated.
Check on the official site:
The ENIGMA criteria are based on a combination of the following: - The 5 class system described for quantitative assessment of variant pathogenicity in Plon et al.(2008) using a multifactorial likelihood model(Goldgar et al., 2004, Easton et al., 2007, Goldgar et al., 2008, Tavtigian et al., 2008) (see Appendix, Table 1); - The 5 class system for interpretation of possible spliceogenic variants and splicing alterations developed by ENIGMA collaborators (Walker et al., 2013); - Generic elements of the 5 class quantitative/qualitative scheme for mismatch repair gene variant classification developed by InSiGHT (Thompson et al., 2014); - Generic elements of the ACMG guidelines for interpretation of sequence variations (Richards et al., 2008); - Classification criteria developed by individual sites participating in ENIGMA, including established country networks; - The classification of sequence changes according to common clinical practice – that is, description of variants generally considered pathogenic (clinically relevant in a genetic counseling setting such that germline variant status is used to inform patient and family management) or non-pathogenic (significant evidence against being a dominant high-risk pathogenic variant).
Furthermore, ENIGMA consortium activity is focused on breast cancer susceptibility genes, not necessarily every DDR genes. If the authors want to illustrate a five classes classification, the ACMG can be sufficient.
Author Response
We thank Reviewers for the positive comments and constructive requests. Reviewer Number 2 considered our manuscript improved. We answered to all the requests of the Reviewer 1 so to improve the quality of our analysis and eliminate inaccuracies. Thank you for your help in improving terminology.
Reviewer 1
The revised version of the manuscript still presents some inaccuracies.
Q1: Line 15: DDR is the acronym for DNA Damage Response (and repair). Check this also throughout the manuscript (e.g. line 46).
A1: Revised “response (and repair) in line 15 and throughout the manuscript)
Q2: Line 19: the genes name has to be written in italics. Check throughout the manuscript.
A2: Revised (all the genes have been written in Italics)
Q3: Line 48 vs Line 53: the abbreviation was just established. Correct line 53.
A3: Revised (“pathogenic variant” in “PV”, line 53)
Q4: Line 65: the paragraph is at risk of to be confounding. The term mutations is used as referred to germline or somatic variants? Germline or somatic are quite specific and variants is a better term instead of mutation.
A4: Revised (“mutations” in “PVs” in line 65).
Q5: Line 181: what does it mean "genetic counseling in the family"? The genetic counseling is offered to the proband/patients. Rephrase e.g genetic counseling for PC patients.
A5: Line 180 has been revised in “Germline analysis is linked to a genetic counseling for PC patients and their relatives”
Q6: Line 189: the ENIGMA criteria are more complex than as stated.
Check on the official site:
The ENIGMA criteria are based on a combination of the following: - The 5 class system described for quantitative assessment of variant pathogenicity in Plon et al.(2008) using a multifactorial likelihood model(Goldgar et al., 2004, Easton et al., 2007, Goldgar et al., 2008, Tavtigian et al., 2008) (see Appendix, Table 1); - The 5 class system for interpretation of possible spliceogenic variants and splicing alterations developed by ENIGMA collaborators (Walker et al., 2013); - Generic elements of the 5 class quantitative/qualitative scheme for mismatch repair gene variant classification developed by InSiGHT (Thompson et al., 2014); - Generic elements of the ACMG guidelines for interpretation of sequence variations (Richards et al., 2008); - Classification criteria developed by individual sites participating in ENIGMA, including established country networks; - The classification of sequence changes according to common clinical practice – that is, description of variants generally considered pathogenic (clinically relevant in a genetic counseling setting such that germline variant status is used to inform patient and family management) or non-pathogenic (significant evidence against being a dominant high-risk pathogenic variant).
Furthermore, ENIGMA consortium activity is focused on breast cancer susceptibility genes, not necessarily every DDR genes. If the authors want to illustrate a five classes classification, the ACMG can be sufficient.
A6: We agree with your comment. ACMG classification can be sufficient for the aim of the manuscript. We replace the sentence as follow “The American College of Medical Genetics and Genomics (ACMG) investigates the clinical significance of sequence variants and creates a guideline for the use of standard terminology [17,18]. According to ACMG criteria, the results of the test can be divided in 5 classes” (lines 188-191).
Reviewer 2 Report
The authors answered and corrected almost all I pointed.
Author Response
We thank Reviewers for the positive comments and constructive requests. Reviewer Number 2 considered our manuscript improved.
REVIEWER 2:
Q1: The authors answered and corrected almost all I pointed.
A1: No more correction pointed by the reviewer
Round 3
Reviewer 1 Report
Line 15 and 47: the acronym DDR means DNA Damage Response (and repair), not Damage DNA response.
Author Response
We thank Reviewer for the comment. We have modified the text, following the suggestions.
REVIEWER 1
Q1: Line 15 and 47: the acronym DDR means DNA Damage Response (and repair), not Damage DNA response.
A1: Revised “Damage DNA response” in “DNA Damage Response (and repair)” , lines 15 and 47.